# From Characters to Words to in Between: Do We Capture Morphology?

## Abstract

Words can be represented by composing the representations of subword units such as word segments, characters, and/or character n-grams. While such representations are effective and may capture the morphological regularities of words, they have not been systematically compared, and it is not understood how they interact with different morphological typologies. On a language modeling task, we present experiments that systematically vary (1) the basic unit of representation, (2) the composition of these representations, and (3) the morphological typology of the language modeled. Our results largely confirm previous findings that character representations are effective across many languages, though we find that a previously unstudied combination of character trigram representations composed with bi-LSTMs outperforms most other settings. However, we also find room for improvement: character models do not match the predictive accuracy of a model with access to explicit morphological analyses.

## 1 Introduction

Continuous representations of words learned by neural networks are central to many NLP tasks (Cho et al., 2014; Chen and Manning, 2014; Dyer et al., 2015). However, directly mapping a finite set of word types to a continuous representation has well-known limitations. First, it makes a closed vocabulary assumption, enabling only generic out-of-vocabulary handling. Second, it cannot exploit systematic functional relationships in learning. For example, *cat* and *cats* stand in the same relationship as *dog* and *dogs*. While this relationship might be discovered for these specific frequent words, it does not help us learn that the same relationship also holds for the much rarer words *tarsier* and *tarsiers*.

These functional relationships reflect the fact that words are composed from smaller units of meaning, or morphemes. For instance, *cats* consists of two morphemes, *cat* and *-s*, with the latter shared by the words *dogs* and *tarsiers*. Modeling this effect is crucial for languages with rich morphology, where vocabulary sizes are larger, many more words are rare, and many more such functional relationships exist. Hence, some models produce word representations as a function of subword units obtained from morphological segmentation or analysis (Luong et al., 2013; Botha and Blunsom, 2014; Cotterell and Schütze, 2015). However, a downside of these models is that they introduce a dependence on morphological segmentation or analysis.

But morphemes typically have similar orthographic representations across words. For example, the morpheme *-s* is realized as *-es* in *finches*, but this variation is limited, and the general relationship between morphology and orthography can be exploited by composing the representations of characters (Ling et al., 2015; Kim et al., 2016), character n-grams (Sperr et al., 2013; **?**; Bojanowski et al., 2016; Botha and Blunsom, 2014), bytes (Plank et al., 2016; Gillick et al., 2016), or combinations thereof (dos Santos and Zadrozny, 2014; Qiu et al., 2014). These models can also represent rare and unknown words, and they produce compact parameterizations. They have the added appeal that they do not depend on morphological analysis, and they raise a provocative question: does NLP benefit from explicit modeling of morphology, or can this be replaced entirely by modeling of characters?

Our understanding of these models is incom-

plete because they have been tested on different tasks, often compared only with direct word embeddings. A number of questions remain open:

1. How do representations based on morphemes compare with those based on characters?
2. What is the best way to compose subword representations?
3. Do character representations adequately substitute for morphological analysis?
4. How do different representations interact with languages of different morphological typologies?

The last question is relevant, since, as Bender (2013) note, languages are typologically diverse, and behaviors exhibited by a model on one language may differ radically on others. Most models implicitly assume concatenative morphology, but typology of many widely-spoken languages is primarily non-concatenative, and it is unclear such models will behave on these languages.

To answer these questions, we performed a systematic comparison across different models for the simple and ubiquitous task of language modeling. We present experiments that vary (1) the type of subword unit; (2) the composition function; and (3) morphological typology. To understand the extent to which character-level models capture true morphological regularities, we present oracle experiments using human morphological annotations instead of automatic morphological segments. Our results show that:

1. For most languages, character-level representations outperform the standard word representations. Most interestingly, a previously unstudied combination of character trigrams composed with bi-LSTMs performs best on the majority of languages.
2. Bi-LSTMs and CNNs are more effective composition functions than addition.
3. Character representations learn functional relationships between orthographically similar words, but are not as accurate as models with explicit knowledge of morphology.
4. Character-level models are effective across a range of morphological typologies, but orthography influences their effectiveness.

## 2  Morphological Typology

A **morpheme** is the smallest unit of meaning in a word. Some morphemes express core meaning (**roots**), while others express one or more depen-

| word | tries |
|---|---|
| morphemes | try+s |
| morphs | tri+es |
| morph. analysis | try+VB+3rd+SG+Pres |

Table 1: The morphemes, morphs, and morphological analysis of *tries*.

dent **features** of the core meaning, such as person, gender, or aspect. A **morphological analysis** identifies the lemma and features of a word. A **morph** is the surface realization of a morpheme (Morley, 2000), which may vary from word to word. These distinctions are shown in Table 1.

Morphological typology classifies languages based on the processes by which morphemes are composed to form words. While most languages will exhibit a variety of such processes, for any given language, some processes are much more frequent than others, and we will broadly identify our experimental languages with these processes.

When morphemes are combined sequentially, the morphology is **concatenative**. However, morphemes can also be composed by **non-concatenative** processes. We consider four broad categories of both concatenative and non-concatenative processes in our experiments.

**Fusional languages** realize multiple features in a single concatenated morpheme. For example, English verbs can express number, person, and tense in a single morpheme:

*wanted* (English)
*want + ed*
*want +* VB+1st+SG+Past

**Agglutinative languages** assign one feature per morpheme. Morphemes are concatenated to form a word and the morpheme boundaries are clear. For example (Haspelmath, 2010):

*okursam* (Turkish)
*oku+r+sa+m*
"read"+AOR+COND+1SG

**Root and Pattern Morphology** forms words by inserting consonants and vowels of dependent morphemes into a consonantal root based on a given pattern. For example, the Arabic root *ktb* ("write") produces (Roark and Sproat, 2007):

*katab* "wrote" (Arabic)
*takaatab* "wrote to each other" (Arabic)

**Reduplication** is a process where a word form is produced by repeating part or all of the root to express new features. For example:

*anak* "child" (Indonesian)
*anak-anak* "children" (Indonesian)
*buah* "fruit" (Indonesian)
*buah-buahan* "various fruits" (Indonesian)

# 3 Representation Models

We compare ten different models, varying subword units and composition functions commonly used in recent work (Table 2). Given word $w$, we compute its representation $\mathbf{w}$ as:

$$\mathbf{w} = f(\mathbf{W}_s, \sigma(w)) \qquad (1)$$

where $\sigma$ is a deterministic function that returns a sequence of subword units; $\mathbf{W}_s$ is a parameter matrix of representations for the vocabulary of subword units; and $f$ is a composition function which takes $\sigma(w)$ and $\mathbf{W}_s$ as input and returns $\mathbf{w}$. All of the representations that we consider take this form, varying only in $f$ and $\sigma$.

## 3.1 Subword Units

We consider four variants of $\sigma$ in Equation 1, each returning a different type of subword unit: character, character trigram, or morph. Morphs are obtained from Morfessor (Smit et al., 2014) or a word segmentation based on Byte Pair Encoding (BPE; Gage (1994)), which has been shown to be effective for handling rare words in neural machine translation (Sennrich et al., 2016). BPE works by iteratively replacing frequent pairs of bytes with a single unused byte. For Morfessor, we use default parameters while for BPE we set the number of merge operations to 10,000.[1] When we segment into character trigrams, we consider all trigrams in the word, including those covering notional beginning and end of word characters, as in Sperr et al. (2013). Example output of $\sigma$ is shown in Table 3.

## 3.2 Composition Functions

We use three variants of $f$ in Eq. 1. The first constructs the representation $\mathbf{w}$ of word $w$ by adding the representations of subwords $s_1, \ldots, s_n = \sigma(w)$, where the representation of $s_i$ is vector $\mathbf{s}_i$.

$$\mathbf{w} = \sum_{i=1}^{n} \mathbf{s}_i \qquad (2)$$

---

[1] BPE takes a single parameter: the number of merge operations. We tried different parameter values (1k, 10k, 100k) and manually examined the resulting segmentation on the English dataset. Qualitatively, 10k gave the most plausible segmentation and we used this setting across all languages.

The only subword unit that we don't compose by addition is characters, since this will produce the same representation for many different words.

Our second composition function is a bidirectional long-short-term memory (**bi-LSTM**), which we adapt based on its use in the character-level model of Ling et al. (2015) and its widespread use in NLP generally. Given $\mathbf{s}_i$ and the previous LSTM hidden state $\mathbf{h}_{i-1}$, an LSTM (Hochreiter and Schmidhuber, 1997) computes the following outputs for the subword at position $i$:

$$\mathbf{h}_i = LSTM(\mathbf{s}_i, \mathbf{h}_{i-1}) \qquad (3)$$
$$\hat{s}_{i+1} = g(\mathbf{V}^T \cdot \mathbf{h}_i) \qquad (4)$$

where $\hat{s}_{i+1}$ is the predicted target subword, $g$ is the softmax function and $\mathbf{V}$ is a weight matrix.

A bi-LSTM (Graves et al., 2005) combines the final state of an LSTM over the input sequence with one over the reversed input sequence. Given the hidden state produced from the final input of the forward LSTM, $\mathbf{h}_n^{fw}$ and the hidden state produced from the final input of the backward LSTM $\mathbf{h}_0^{bw}$, we compute the word representation as:

$$\mathbf{w}_t = \mathbf{W}_f \cdot \mathbf{h}_n^{fw} + \mathbf{W}_b \cdot \mathbf{h}_0^{bw} + \mathbf{b} \qquad (5)$$

where $\mathbf{W}_f$ and $\mathbf{W}_b$ are parameter matrices and $\mathbf{h}_n^{fw}$ and $\mathbf{h}_0^{bw}$ are forward and backward LSTM states.

The third composition function is a convolutional neural network (**CNN**) with highway layers, as in Kim et al. (2016). Let $c_1, \ldots, c_k$ be the sequence of characters of word $w$. The character embedding matrix is $\mathbf{C} \in \mathbb{R}^{d \times k}$, where the $i$-th column corresponds to the embeddings of $c_i$. We first apply a narrow convolution between $\mathbf{C}$ and a filter $\mathbf{F} \in \mathbb{R}^{d \times n}$ of width $n$ to obtain a feature map $\mathbf{f} \in \mathbf{R}^{k-n+1}$. In particular, the computation of the $j$-th element of $\mathbf{f}$ is defined as

$$\mathbf{f}[j] = tanh(\langle \mathbf{C}[*, j : j + n - 1], \mathbf{F}\rangle + b) \qquad (6)$$

where $\langle A, B\rangle = \mathtt{Tr}(\mathbf{A}\mathbf{B}^T)$ is the Frobenius inner product and $b$ is a bias. The CNN model applies filters of varying width, representing features of character n-grams. We then calculate the max-over-time of each feature map.

$$y_j = \max_j \mathbf{f}[j] \qquad (7)$$

and concatenate them to derive the word representation $\mathbf{w}_t = [y_1, \ldots, y_m]$, where $m$ is the number of filters applied. Highway layers allow some dimensions of $\mathbf{w}_t$ to be carried or transformed. Since it can learn character n-grams directly, we only use the CNN with character input.

| Models | Subword Unit(s) | Composition Function |
|---|---|---|
| Sperr et al. (2013) | words, character n-grams | addition |
| Luong et al. (2013) | morphs (Morfessor) | recursive NN |
| Botha and Blunsom (2014) | words, morphs (Morfessor) | addition |
| Qiu et al. (2014) | words, morphs (Morfessor) | addition |
| dos Santos and Zadrozny (2014) | words, characters | CNN |
| Cotterell and Schütze (2015) | words, morphological analyses | addition |
| Sennrich et al. (2016) | morphs (BPE) | none |
| Kim et al. (2016) | characters | CNN |
| Ling et al. (2015) | characters | bi-LSTM |
| Wieting et al. (2016) | character n-grams | addition |
| Bojanowski et al. (2016) | character n-grams | addition |
| Vylomova et al. (2016) | characters, morphs (Morfessor) | bi-LSTM, CNN |
| Miyamoto and Cho (2016) | words, characters | bi-LSTM |
| Lee et al. (2016) | characters | CNN |

Table 2: Summary of previous work on representing words through compositions of subword units.

| Unit | Output of $\sigma(\textbf{\textit{wants}})$ |
|---|---|
| Morfessor | ˆwant, s$ |
| BPE | ˆw, ants$ |
| char-trigram | ˆwa, wan, ant, ts$ |
| character | ˆ, w, a, n, t, s, $ |
| analysis | want+VB, +3rd, +SG, +Pres |

Table 3: Input representations for *wants*

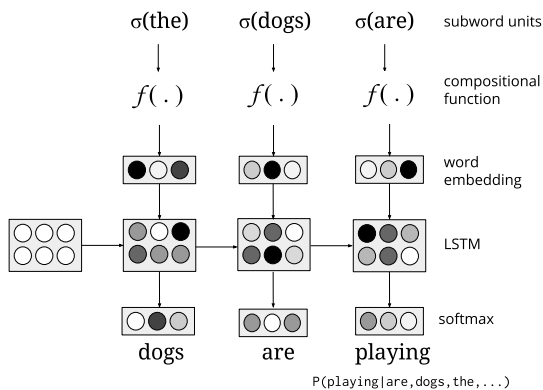

Figure 1: Our LSTM-LM architecture.

### 3.3 Language Model

We use language models (LM) because they are simple and fundamental to many NLP applications. Given a sequence of text $s = w_1, \ldots, w_T$, our LM computes the probability of $s$ as:

$$P(w_1, \ldots, w_T) = \prod_{t=1}^{T} P(y_t | w_1, \ldots, w_{t-1}) \quad (8)$$

where $y_t = w_t$ if $w_t$ is in the output vocabulary and $y_t = \text{UNK}$ otherwise.

Our language model is an LSTM variant of recurrent neural network language (RNN) LM (Mikolov et al., 2010). At time step $t$, it receives input $w_t$ and predicts $y_{t+1}$. Using Eq. 1, it first computes representation $\mathbf{w}_t$ of $w_t$. Given this representation and previous state $\mathbf{h}_{t-1}$, it produces a new state $\mathbf{h}_t$ and predicts $y_{t+1}$:

$$\mathbf{h}_t = LSTM(\mathbf{w}_t, \mathbf{h}_{t-1}) \quad (9)$$
$$\hat{y}_{t+1} = g(\mathbf{V}^T \cdot \mathbf{h}_t) \quad (10)$$

where $g$ is a softmax function over the vocabulary yielding the probability in Equation 8. Note that this design means that we can *predict* only words from a fixed output vocabulary, so our models differ only in their representation of context words. This makes design it possible to compare language models using perplexity on a fixed output space, though open vocabulary word prediction is an interesting direction for future work.

The complete architecture of our system is shown in Figure 1, showing segmentation function $\sigma$ and composition function $f$ from Equation 1.

## 4 Experiments

We perform experiments on ten languages (Table 4). We use datasets from Ling et al. (2015) for English and Turkish. For Czech and Russian we use Universal Dependencies (UD) v1.3 (Nivre et al., 2015). For other languages, we use prepro-

| Typology | Languages | #tokens | #types |
|---|---|---|---|
| Fusional | Czech | 1.2M | 125.4K |
| | English | 1.2M | 81.1K |
| | Russian | 0.8M | 103.5K |
| Agglutinative | Finnish | 1.2M | 188.4K |
| | Japanese | 1.2M | 59.2K |
| | Turkish | 0.6M | 126.2K |
| Root&Pattern | Arabic | 1.4M | 137.5K |
| | Hebrew | 1.1M | 104.9K |
| Reduplication | Indonesian | 1.2M | 76.5K |
| | Malaysian | 1.2M | 77.7K |

Table 4: Statistics of our datasets.

cessed Wikipedia data (Al-Rfou et al., 2013).[2] For each dataset, we use approximately 1.2M tokens to train, and approximately 150K tokens each for development and testing. Preprocessing involves lowercasing (except for character models) and removing hyperlinks.

To ensure that we compared models and not implementations, we reimplemented all models in a single framework using Tensorflow (Abadi et al., 2015). We use a common setup for all experiments based on that of (Ling et al., 2015; Kim et al., 2016; Miyamoto and Cho, 2016). In preliminary experiments, we confirmed that our models produced similar patterns of perplexities for the reimplemented word and character LSTM models of Ling et al. (2015). Even following detailed discussion with Ling (p.c.), we were unable to reproduce their perplexities exactly—our English reimplementation gives lower perplexities; our Turkish higher—but we do reproduce their general result that character bi-LSTMs outperform word models. We suspect that different preprocessing and the stochastic learning explains differences in perplexities. Our final model with bi-LSTMs composition follows Miyamoto and Cho (2016) as it gives us the same results reported in their paper.

### 4.1 Training and Evaluation

Our LSTM-LM uses two hidden layers with 200 hidden units and representation vectors for words, characters, and morphs all have dimension 200. All parameters are initialized uniformly at random from -0.1 to 0.1, trained by stochastic gradient de-

[2] The Arabic and Hebrew dataset are unvocalized. Japanese mixes Kanji, Katakana, Hiragana, and Latin characters (for foreign words). Hence, a Japanese character can correspond to a character, syllable, or word. The preprocessed dataset is already word-segmented.

scent with mini-batch size of 32, time steps of 20, for 50 epochs. To avoid overfitting, we apply dropout with probability 0.5. For all models which do not use bi-LSTM composition, we start with a learning rate of 1.0 and decrease it by half if the validation perplexity does not decrease by 0.1 after 3 epochs. For models with bi-LSTMs composition, we use a constant learning rate of 0.2 and stop training when validation perplexity does not improve after 3 epochs. For the character CNN model, we use the same settings as the *small model* of Kim et al. (2016).

To make our results comparable to Ling et al. (2015), for each language we limit the output vocabulary to the most frequent 5,000 training words plus an unknown word token. To learn to predict unknown words, we follow Ling et al. (2015): in training, words that occur only once are stochastically replaced with the unknown token with probability 0.5. To evaluate the models, we compute perplexity on the test data.

## 5 Results and Analysis

Table 5 presents our main results. In six of ten languages, character-trigram representations composed with bi-LSTMs achieve the lowest perplexities. As far as we know, this particular model has not been tested before, though it is similar (but more general) than the model of Sperr et al. (2013). We can see that the performance of character, character trigrams, and BPE are very competitive. Composition by bi-LSTMs or CNN is more effective than addition, except for Turkish. We also observe that BPE always outperforms Morfessor, even for the agglutinative languages. We now turn to a more detailed analysis by morphological typology.

**Fusional languages.** For these languages, character trigrams composed with bi-LSTMs outperformed all other models, particularly for Czech and Russian (up to 20%), which is unsurprising since both are morphologically richer than English.

**Agglutinative languages.** We observe different results for each language. For Finnish, character trigrams composed with bi-LSTMs achieves the best perplexity. Surprisingly, for Turkish character trigrams composed via addition is best and addition also performs quite well for other representations, potentially useful since the addition function is simpler and faster than bi-LSTMs. We

| Language | word | character | | char trigrams | | BPE | | Morfessor | | %imp |
|---|---|---|---|---|---|---|---|---|---|---|
| | | bi-lstm | CNN | add | bi-lstm | add | bi-lstm | add | bi-lstm | |
| Czech | 41.46 | 34.25 | 36.60 | 42.73 | **33.59** | 49.96 | 33.74 | 47.74 | 36.87 | 18.98 |
| English | 46.40 | 43.53 | 44.67 | 45.41 | **42.97** | 47.51 | 43.30 | 49.72 | 49.72 | 7.39 |
| Russian | 34.93 | 28.44 | 29.47 | 35.15 | **27.72** | 40.10 | 28.52 | 39.60 | 31.31 | 20.64 |
| Finnish | 24.21 | 20.05 | 20.29 | 24.89 | **18.62** | 26.77 | 19.08 | 27.79 | 22.45 | 23.09 |
| Japanese | 98.14 | 98.14 | **91.63** | 101.99 | 101.09 | 126.53 | 96.80 | 111.97 | 99.23 | 6.63 |
| Turkish | 66.97 | 54.46 | 55.07 | **50.07** | 54.23 | 59.49 | 57.32 | 62.20 | 62.70 | 25.24 |
| Arabic | 48.20 | 42.02 | 43.17 | 50.85 | **39.87** | 50.85 | 42.79 | 52.88 | 45.46 | 17.28 |
| Hebrew | 38.23 | 31.63 | 33.19 | 39.67 | **30.40** | 44.15 | 32.91 | 44.94 | 34.28 | 20.48 |
| Indonesian | 46.07 | 45.47 | 46.60 | 58.51 | 45.96 | 59.17 | **43.37** | 59.33 | 44.86 | 5.86 |
| Malay | 54.67 | 53.01 | **50.56** | 68.51 | 50.74 | 68.99 | 51.21 | 68.20 | 52.50 | 7.52 |

Table 5: Language model perplexities on test, showing improvement of the best system over words.

suspect that this is due to the fact that Turkish morphemes are reasonably short, hence well-approximated by character trigrams. For Japanese, we observe that the improvements from character representation more modest than in other language.

**Root and Pattern.** For these languages, character trigrams composed with bi-LSTMs also achieve the best perplexity. We had wondered with CNNs would be more effective for root-and-patter morphology, but since these data are unvocalized, it is more likely that non-concatenative effects are minimized, though we do still find morphological variants with consonantal inflections that behave more like concatenation. For example, *maktab* (root:*ktb*) is written as *mktb*. We suspect this makes character trigrams quite effective since they match the tri-consonantal root patterns among words which share the same root.

**Reduplication.** For Indonesian, BPE morphs composed with bi-LSTMs model obtain the best perplexity. For Malay, the character CNN outperforms other models. However, these improvements are small compared to other languages. This likely reflects that Indonesian and Malay are only moderately inflected, where inflection involves both concatenative and non-concatenative processes.

## 5.1 Effects of Morphological Analysis

In the experiments above, we used unsupervised morphological *segmentation* as a proxy for morphological *analysis* (Table 3). However, as discussed in Section 2, this is quite approximate, so it is natural to wonder what would happen if we had an ideal morphological analysis. If character-

| Languages | Addition | bi-LSTM |
|---|---|---|
| Czech | 51.8 | **30.07** |
| Russian | 41.82 | **26.44** |

Table 6: Perplexity results using hand-annotated morphological analyses (cf. Table 5).

level models were adequate to model the effects of morphology, then they would have similar predictive accuracy. To answer this question, we used the human-annotated morphological analyses provided in the UD datasets for Czech and Russian, the only languages in our set for which these analyses were available. In these experiments we treat the lemma and each morphological feature as a subword unit.

The results (Table 6) show that bi-LSTM composition of these representations outperforms all other models for both languages. These results demonstrate that neither character representations nor unsupervised segmentation is a perfect replacement for manual morphological analysis, at least in terms of predictive accuracy. Especially in light of character-level results, they also imply that current methods of unsupervised morphological analysis are inadequate substitutes for morphological analysis.

## 5.2 Automatic Morphological Analysis

The oracle experiments show promising results if we have annotated data. But these annotations are expensive, so investigated the effects of automatic morphological analysis. We obtained analyses for Arabic with the MADAMIRA (Pasha et al., 2014). As in the experiment using annotations, we treated each morphological feature as a sub-

word unit. The resulting perplexities of **71.94** and **42.85** for addition and bi-LSTMs, respectively, are worse than those obtained with character trigrams (**39.87**), though it approaches the best perplexities.

## 5.3 Targeted Perplexity Results

A difficulty in interpreting the results of Table 5 with respect to specific morphological processes is that perplexity is measured for all words. But these processes do not apply to all words, so it may be that the effects of specific morphological processes are washed out. To get a clearer picture, we measured perplexity for only specific subsets of words in our test data: specifically, given target word $w_i$, we measure perplexity of word $w_{i+1}$. In other words, we analyze the perplexities *when the inflected words of interest are in the most recent history*, exploiting the recency bias of our LSTM-LM. This is the perplexity most likely to be strongly affected by different representations, since we do not vary representations of the predicted word itself.

We look at several cases: nouns and verbs in Czech and Russian, where word classes can be identified from annotations, and reduplication in Indonesian, which we can identify mostly automatically. For each analysis, we also distinguish between *frequent* cases, where the inflected word occurs more than ten times in the training data, and *rare* cases, where it occurs fewer than ten times. We compare only bi-LSTM models.

For Czech and Russian, we again use the UD annotation to identify words of interest. The results (Table 7), show that manual morphological analysis uniformly outperforms other subword models, with an especially strong effect for Czech nouns, suggesting that other models do not capture useful predictive properties of a morphological analysis. We do however note that character trigrams achieve low perplexities in most cases, similar to overall results (Table 5). We also observe that the subword models are more effective for rare verbs.

For Indonesian, we exploit the fact that the hyphen symbol '-' typically separates the first and second occurrence of a reduplicated morpheme, as in the examples of Section 2. We use the presence of word tokens containing hyphens to estimate the percentage of those exhibiting reduplication. As shown in Table 8, the numbers are quite low.

Table 9 shows results for reduplication. In

| Inflection | Model | all | frequent | rare |
|---|---|---|---|---|
| Czech nouns | word | 61.21 | 56.84 | 72.96 |
| | characters | 51.01 | 47.94 | 59.01 |
| | char-trigrams | 50.34 | 48.05 | 56.13 |
| | BPE | 53.38 | 49.96 | 62.81 |
| | morph. analysis | **40.86** | **40.08** | **42.64** |
| Czech verbs | word | 81.37 | 74.29 | 99.40 |
| | characters | 70.75 | 68.07 | 77.11 |
| | char-trigrams | 65.77 | 63.71 | 70.58 |
| | BPE | 74.18 | 72.45 | 78.25 |
| | morph. analysis | **59.48** | **58.56** | **61.78** |
| Russian nouns | word | 45.11 | 41.88 | 48.26 |
| | characters | 37.90 | 37.52 | 38.25 |
| | char-trigrams | 36.32 | 34.19 | 38.40 |
| | BPE | 43.57 | 43.67 | 43.47 |
| | morph. analysis | **31.38** | **31.30** | **31.50** |
| Russian verbs | word | 56.45 | 47.65 | 69.46 |
| | characters | 45.00 | 40.86 | 50.60 |
| | char-trigrams | 42.55 | 39.05 | 47.17 |
| | BPE | 54.58 | 47.81 | 64.12 |
| | morph. analysis | **41.31** | **39.8** | **43.18** |

Table 7: Average perplexities of words that occur after nouns and verbs.

| Language | type-level (%) | token-level (%) |
|---|---|---|
| Indonesian | 1.10 | 2.60 |
| Malay | 1.29 | 2.89 |

Table 8: Percentage of full reduplication on the type and token level.

| Model | all | frequent | rare |
|---|---|---|---|
| word | 101.71 | 91.71 | 156.98 |
| characters | **99.21** | **91.35** | **137.42** |
| BPE | 117.2 | 108.86 | 156.81 |

Table 9: Average perplexities of words that occur after reduplicated words in the test set.

contrast with the overall results, BPE bi-LSTMs model produce the worse perplexities, while character bi-LSTMs produce the best, suggesting that these models are more effective for reduplication.

Looking more closely at BPE segmentation of reduplicated words, we found that only 6 of 252 reduplicated words have a correct word segmentation, with the reduplicated morpheme often being combined differently with the notional start-of-word or hyphen character. One the other hand BPE correctly learns 8 out of 9 Indonesian prefixes and 4 out of 7 Indonesian suffixes.[3] This analysis supports our intuition that the improvement from BPE might come from its modeling of concatenative morphology.

---

[3] We use Indonesian preprefixes, prefixes and suffixes listed in Larasati et al. (2011)

| Model | Frequent Words | | | Rare Words | | OOV words | |
|---|---|---|---|---|---|---|---|
| | *man* | *including* | *relatively* | *unconditional* | *hydroplane* | *uploading* | *foodism* |
| word | person | like | extremely | nazi | molybdenum | - | - |
| | anyone | featuring | making | fairly | your | - | - |
| | children | include | very | joints | imperial | - | - |
| | men | includes | quite | supreme | intervene | - | - |
| BPE LSTM | ii | called | newly | unintentional | emphasize | upbeat | vigilantism |
| | hill | involve | never | ungenerous | heartbeat | uprising | pyrethrum |
| | text | like | essentially | unanimous | hybridized | handling | pausanias |
| | netherlands | creating | least | unpalatable | unplatable | hand-colored | footway |
| char-trigrams LSTM | mak | include | resolutely | unconstitutional | selenocysteine | drifted | tuaregs |
| | vill | includes | regeneratively | constitutional | guerrillas | affected | quft |
| | cow | undermining | reproductively | unimolecular | scrofula | conflicted | subjectivism |
| | maga | under | commonly | medicinal | seleucia | convicted | tune-up |
| char-LSTM | mayr | inclusion | relates | undamaged | hydrolyzed | musagte | formulas |
| | many | insularity | replicate | unmyelinated | hydraulics | mutualism | formally |
| | mary | includes | relativity | unconditionally | hysterotomy | mutualists | fecal |
| | may | include | gravestones | uncoordinated | hydraulic | meursault | foreland |
| char-CNN | mtn | include | legislatively | unconventional | hydroxyproline | unloading | fordism |
| | mann | includes | lovely | unintentional | hydrate | loading | dadaism |
| | jan | excluding | creatively | unconstitutional | hydrangea | upgrading | popism |
| | nun | included | negatively | untraditional | hyena | upholding | endemism |

Table 10: Top nearest neighbours (cosine similarities) of semantically and syntactically similar words.

| Query | Top nearest neighbours |
|---|---|
| kota-kota (*cities*) | wilayah-wilayah (*areas*), pulau-pulau (*islands*), negara-negara (*countries*), bahasa-bahasa (*languages*), koloni-koloni (*colonies*) |
| berlembah-lembah (*have many valleys*) | berargumentasi (*argue*), bercakap-cakap (*converse*), berkemauan (*will*), berimplikasi (*imply*), berketebalan (*have a thickness*) |

Table 11: Top nearest neighbours (cosine similarities) of Indonesian reduplicated words.

## 5.4 Qualitative Analysis

Table 10 presents nearest neighbors under cosine similarity for in-vocabulary, rare, and out-of-vocabulary (OOV) words.[4] For frequent words, standard word embeddings are clearly superior for lexical meaning. Character and morph representations tend to find words that are orthographically similar, suggesting that they better at modeling dependent than root morphemes. The same pattern holds for rare and OOV words. We suspect that the subword models outperform words on language modeling because they exploit affixes to signal word class. We also noticed similar patterns in Japanese.

We analyze reduplication by querying reduplicated words to find their nearest neighbors using the BPE bi-LSTM model. If the model were sensitive to reduplication, we would expect to see the morphological variants word in the top-n nearest neighbors. However, from Table 11, this is not so. With the partially reduplicated query *berlembah-lembah*, we do not find the lemma *lembah*.

---

[4]https://radimrehurek.com/gensim/

## 6 Conclusion

We presented a systematic comparison of word representation models with different levels of morphological awareness, across languages with different morphological typologies. Our results confirm previous findings that character-level models are effective for many languages, but these models do not match the predictive accuracy of model with explicit knowledge of morphology, and our qualitative analysis suggests that they learn orthographic similarity of affixes, and lose the meaning of root morphemes.

Although morphological analyses are available in limited quantities, our results suggest that there might be utility in semi-supervised learning from partially annotated data. Across languages with different typologies, our experiments show that the subword unit models are most effective on agglutinative languages. However, these results do not generalize to all languages, since factors such as morphology and orthography affect the utility of these representations. We plan to explore these effects in future work.

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
