# Peer review of "From Characters to Words to in Between: Do We Capture Morphology?"

_ACL 2017 — decision unknown_

[Official Review · Reviewer 1 · rating 3 · confidence 3]
soundness 5 · originality 5 · clarity 3 · impact 3 · substance 3 · appropriateness 5 · meaningful comparison 3 · presentation format Poster

- Strengths:
i. Motivation is well described.
ii. Provides detailed comparisons with various models across diverse languages

- Weaknesses:
i.          The conclusion is biased by the selected languages. 
ii.           The experiments do not cover the claim of this paper completely.

- General Discussion:
This paper issues a simple but fundamental question about word representation:
what subunit of a word is suitable to represent morphologies and how to compose
the units. To answer this question, this paper applied word representations
with various subunits (characters, character-trigram, and morphs) and
composition functions (LSTM, CNN, and a simple addition) to the language
modeling task to find the best combination. In addition, this paper evaluated
the task for more than 10 languages. This is because languages are
typologically diverse and the results can be different according to the word
representation and composition function. From their experimental results, this
paper concluded that character-level representations are more effective, but
they are still imperfective in comparing them with a model with explicit
knowledge of morphology. Another conclusion is that character-trigrams show
reliable perplexity in the majority of the languages. 

However, this paper leaves some issues behind.
-         First of all, there could be some selection bias of the experimental
languages. This paper chose ten languages in four categories (up to three
languages per a category). But, one basic question with the languages is “how
can it be claimed that the languages are representatives of each category?”
All the languages in the same category have the same tendency of word
representation and composition function? How can it be proved? For instance,
even in this paper, two languages belonging to the same typology
(agglutinative) show different results. Therefore, at least to me, it seems to
be better to focus on the languages tested in this paper instead of drawing a
general conclusions about all languages. 
-         There is some gap between the claim and the experiments. Is the
language modeling the best task to prove the claim of this paper? Isn’t there
any chance that the claim of this paper breaks in other tasks? Further
explanation on this issue is needed.
-         In Section 5.2, this paper evaluated the proposed method only for
Arabic. Is there any reason why the experiment is performed only for Arabic?
There are plenty of languages with automatic morphological analyzers such as
Japanese and Turkish.
-         This paper considers only character-trigram among various n-grams. Is
there any good reason to choose only character-trigram? Is it always better
than character-bigram or character-fourgram? In general, language modeling with
n-grams is affected by corpus size and some other factors. 

Minor typos: 
- There is a missing reference in Introduction. (88 line in Page 1)
- root-and-patter -> root-and-pattern (524 line in Page 6)

[Official Review · Reviewer 2 · rating 4 · confidence 4]
soundness 5 · originality 5 · clarity 5 · impact 3 · substance 4 · appropriateness 5 · meaningful comparison 3 · presentation format Oral Presentation

tldr: The authors compare a wide variety of approaches towards sub-word
modelling in language modelling, and show that modelling morphology gives the
best results over modelling pure characters. Further, the authors do some
precision experiments to show that the biggest benefit towards sub-word
modelling is gained after words typically exhibiting rich morphology (nouns and
verbs). The paper is comprehensive and the experiments justify the core claims
of the paper. 

- Strengths:

1) A comprehensive overview of different approaches and architectures towards
sub-word level modelling, with numerous experiments designed to support the
core claim that the best results come from modelling morphemes.

2) The authors introduce a novel form of sub-word modelling based on character
tri-grams and show it outperforms traditional approaches on a wide variety of
languages.

3) Splitting the languages examined by typology and examining the effects of
the models on various typologies is a welcome introduction of linguistics into
the world of language modelling.

4) The analysis of perplexity reduction after various classes of words in
Russian and Czech is particularly illuminating, showing how character-level and
morpheme-level models handle rare words much more gracefully. In light of these
results, could the authors say something about how much language modelling
requires understanding of semantics, and how much it requires just knowing
various morphosyntactic effects?

- Weaknesses:

1) The character tri-gram LSTM seems a little unmotivated. Did the authors try
other character n-grams as well? As a reviewer, I can guess that character
tri-grams roughly correspond to morphemes, especially in Semitic languages, but
what made the authors report results for 3-grams as opposed to 2- or 4-? In
addition, there are roughly 26^3=17576 possible distinct trigrams in the Latin
lower-case alphabet, which is enough to almost constitute a word embedding
table. Did the authors only consider observed trigrams? How many distinct
observed trigrams were there?

2) I don't think you can meaningfully claim to be examining the effectiveness
of character-level models on root-and-pattern morphology if your dataset is
unvocalised and thus doesn't have the 'pattern' bit of 'root-and-pattern'. I
appreciate that finding transcribed Arabic and Hebrew with vowels may be
challenging, but it's half of the typology.

3) Reduplication seems to be a different kind of phenomenon to the other three,
which are more strictly morphological typologies. Indonesian and Malay also
exhibit various word affixes, which can be used on top of reduplication, which
is a more lexical process. I'm not sure splitting it out from the other
linguistic typologies is justified.

- General Discussion:

1) The paper was structured very clearly and was very easy to read.

2) I'm a bit puzzled about why the authors chose to use 200 dimensional
character embeddings. Once the dimensionality of the embedding is greater than
the size of the vocabulary (here the number of characters in the alphabet),
surely you're not getting anything extra?

-------------------------------

Having read the author response, my opinions have altered little. I still think
the same strengths and weakness that I have already discussed hold.